# Towards Automatic Detection of Precipitates in Inconel 625 Superalloy Additively Manufactured by the L-PBF Method

**DOI:** 10.3390/ma14164507

**Published:** 2021-08-11

**Authors:** Piotr Macioł, Jan Falkus, Paulina Indyka, Beata Dubiel

**Affiliations:** 1Faculty of Metals Engineering and Industrial Computer Science, AGH University of Science and Technology, Czarnowiejska 66, 30-054 Kraków, Poland; jfalkus@agh.edu.pl (J.F.); bdubiel@agh.edu.pl (B.D.); 2Solaris National Synchrotron Radiation Centre, Faculty of Chemistry, Jagiellonian University, Czerwone Maki 98, 30-392 Kraków, Poland; paulina.indyka@uj.edu.pl

**Keywords:** inconel 625, additive manufacturing, EDS microanalysis, automatic image analysis

## Abstract

In our study, the comparison of the automatically detected precipitates in L-PBF Inconel 625, with experimentally detected phases and with the results of the thermodynamic modeling was used to test their compliance. The combination of the complementary electron microscopy techniques with the microanalysis of chemical composition allowed us to examine the structure and chemical composition of related features. The possibility of automatic detection and identification of precipitated phases based on the STEM-EDS data was presented and discussed. The automatic segmentation of images and identifying of distinguishing regions are based on the processing of STEM-EDS data as multispectral images. Image processing methods and statistical tools are applied to maximize an information gain from data with low signal-to-noise ratio, keeping human interactions on a minimal level. The proposed algorithm allowed for automatic detection of precipitates and identification of interesting regions in the Inconel 625, while significantly reducing the processing time with acceptable quality of results.

## 1. Introduction

Recent advances in additive manufacturing (AM) of metals and alloys require intensification of the research of their microstructure, which should be precisely controlled to assure the good quality of the manufactured products. Moreover, the evolution of the microstructure in the working conditions should be recognized. It is especially important due to the growing application of AM technology to produce parts made of superalloys for aerospace, energy, automotive, and chemical industries. From the perspective of materials science, significant challenges remain in the field of understanding the effect of the build process from metal powder experiencing rapid melting and solidification on the microstructure and its development during high temperature exposure. 

Due to the elimination of difficulties with subtractive machining and the possibility of making parts with complex geometry, the additive manufacturing of nickel-based superalloy components by laser powder fusion (L-PBF) has recently attracted great interest [1]. One of the superalloys most widely applied for L-PBF manufacturing is Inconel 625, which exhibits high temperature strength and exceptional corrosion resistance in harmful environments. High cooling rate and repeated heating and cooling cycles during the L-PBF process shape a distinctly different microstructure of Inconel 625 than in other methods. In as-built condition, it consists of paths resulting from the bonding of powder particles by the laser beam, in which the rapidly solidified melt pools are distinguished. Inside them, fine grains with a very fine cellular-dendritic structure are present. The non-equilibrium solidification conditions promote the microsegregation of alloying elements and formation of nano-precipitates, as well as the substructure of dislocation cells [2,3,4,5,6,7].

One of the important phenomena that require careful study is the examination of the precipitates present in L-PBF Inconel 625 and their evolution after prolonged exposure at high temperature. 

### 1.1. Microstructural and Compositional Characterization of Precipitates in Additively Manufactured Inconel 625 Superalloy

The literature data concerning nano-sized precipitates in L-PBF Inconel 625 are mostly related to as-built and stress-relief annealed condition [2,3,4,5,6], in which the occurrence of the γ″ phase and δ phase, as well as MC, M_6_C, and M_23_C_6_ carbides, was established. The presence of the Laves phase and P phase is also reported [8]. In addition, nanometric oxide inclusions, which are formed by the oxidation of the feed powder and/or phenomena occurring during additive manufacturing, are observed [9,10]. These particles were detected on the basis of different methods, such as SEM and TEM imaging, X-ray, and electron diffraction, as well as EDS microanalysis of chemical composition. However, there is limited data concerning the evolution of precipitates after high temperature exposure of L-PBF Inconel 625. Examination of the evolution of the microstructure and the local changes in chemical composition caused by annealing can be performed by scanning transmission electron microscopy (STEM) imaging, combined with energy dispersive X-ray (EDS) elemental mapping, which allows for both microstructure observation and determination of elemental distribution within nanoareas. This technique requires the use of thin specimens transparent to the electron beam. In the STEM mode, a focused electron beam scans the sample. Due to the interaction of electrons with the sample, different signals are excited and collected at each scanned point by appropriate detectors. The collected signals can be used to create various types of images. High-angle annular dark-field (HAADF) images show the intensity distribution of electrons scattered near the atomic nuclei of the atoms in the sample. The intensity of the images depends on the square of the atomic number (Z) and the thickness of the sample, so a mass-thickness contrast is obtained. The contrast in STEM-HAADF mode is also influenced by the deformation fields around structural defects, e.g., dislocations [11]. In turn, EDS spectral images, showing maps of the distribution of chemical elements, are obtained by collecting EDS spectra at each point of the scanned sample (Figure 1). The information obtained from the compositional maps is influenced by many factors related to the instrument configuration, the specimen, and the settings of the data analysis software [12].

### 1.2. Automatic Detection of Particles in Metal Alloys Produced by Additive Manufacturing

Interpretation of micrographs, including EDS spectral images, is a tedious and time-consuming work. It is even more challenging if the results do not offer high quality (meaning sufficient number of X-ray counts per pixel). Analyzing of acquired data requires precise processing by an experience human researcher. Since this work is frequently based on trial-and-error approach, mainly, while adjusting the background level, it is not only time-consuming but may lead to incorrect or only partially correct results. Automatic analysis of images cannot replace a skilled human researcher (at least not today) but might save a lot of most cumbersome work and provide a good starting point for in-depth manual investigation. 

There are several domains, where automatic image/data processing could be useful, from the most typical (denoising) to the ones requiring domain-specific knowledge, such as supporting phases identification. The phases’ identification is in fact a form of clustering of data, and many types of data science are applicable.

### 1.3. Denoising

STEM imaging and EDS mapping require considerable time to acquire a single image. The need of reducing that time introduces artifacts, which are generally described as a noise. Many techniques have been developed to deal with this issue. Some are general methods of image processing; others are focused on processing microscope images. Often, simple general purpose filters, such as a Gaussian blur, median filter, frequency space low-pass, or Wiener filters, are applied [13,14,15]. More advanced techniques consider specificity of imaged materials character/features. In metal alloys, a periodicity of a structure is usually used. An example of employing this approach for high-resolution STEM is presented, i.e., in Reference [16]. The most successful image denoising algorithm employing recognition of similar features rely on non-local detection and averaging of self-similar image regions. The first algorithm based on this strategy is the non-local means filter (NLM) in Reference [17]. Due to the richness in self-similarity of electron micrographs of crystals, NLM is, in principle, very well suited for denoising such micrographs [18,19]. Recently, Mevenkamp et al. [20] proposed a multi-modal and multi-scale non-local means (M3S-NLM) method to extract atomically resolved spectroscopic maps. Their approach joins NLM approach with a multimodality based on searching on similarities not only between different features recorded with a single method and spread spatially on image but also considering several bound signals. The method is limited to periodic structures. Results presented by Mevenkamp et al. are based on a synthetic structure, generated from a single real image. Such approaches based on a cyclicity and NLM works very well if an imaged object exhibits cyclic structure or at least features on the image are strongly similar. However, none of these requirements is fulfilled in case of imaging of cellular microstructure in L-PBF Inconel 625. 

The most common, classical, and widely used tools for automatic denoising are binning and smearing methods. The binning is based on replacing of a set (usually a rectangle or a square) of pixels with a single one, with a signal being a sum of signals from replaced pixels. The main disadvantage of this approach is significant loss of spatial resolution. The smearing is based on averaging (unweighted, linear or nonlinear weighted) of adjacent pixels. This approach reduces noise and might be very efficient; however, it usually requires manual fitting of weighting function parameters. Improperly chosen values lead to ineffective denoising or losing important information. Since classical denoising methods have reduced application in automatic processing of highly noised and non-uniform images, recently, in material science, most attention is paid to Machine Learning techniques. Ede presented a neural network trained to remove noise from micrographs. The presented case studies show significant improve of image quality [21]. The author claims that once trained network can be successfully used for a wide range of types of images, including STEM images. However, the level of noise present on the exemplary images was never high enough to hinder automatic identification of features. Other examples of ML algorithms for noise removal in microscopy are presented in References [22,23]. 

While ML techniques can be effective, they are burdened with two important drawbacks. At first, the most of ML techniques is based on a black-box approach. Hence, while it is possible to get an answer for the question of what is the prediction, it is not possible to explain the prediction (black-box approach). Second, ML techniques require manually labeled data (supervised ML) or huge datasets (unsupervised ML). Due to those important issues, other techniques useful in automatic analysis of micrographs are also investigated. One of the most promising is Principal Component Analysis (PCA). Potapov et al. and Potapov and Lubk [24,25] presented an application of PCA to enhance quality of highly noised EDS images. Combination of several classical filtering methods with PCA are presented. Processed images have much better quality, but several manually fitted coefficients were necessary. 

Some reviews describing various advanced denoising methods of microscope images (e.g., modified Rudin-Osher-Fatemi model, the adaptive Total variation method or Poisson Unbiased Risk Estimation—Linear Expansion of Thresholds) might be found also in References [26,27].

### 1.4. Segmentation

Other relatively common use of image processing techniques in micrographs’ analysis is segmentation. One of the primary purposes of microstructural analysis of metal alloys is examination of grain size and morphology, as well as identification of structural compounds, phases, and other features. Traditionally, this job is performed by a human researcher on the ground of his knowledge of investigated material and with the use of microscopy techniques. If a metal alloy is fabricated using additive manufacturing, this task gets somewhat complicated because of a combinatorial explosion of possible microstructural features to identify. Hence, any support which might be provided by computational techniques is very welcome. The first step in computer aided image analysis is distinguishing of present features. The problem of segmentation of images is well known and important in many disciplines, such as in i.a. autonomous driving. A review of segmentation techniques for general applications including traditional and modern ML-based approaches might be found in Reference [28]. 

Segmentation of microscope images has been a subject of many publications [29,30,31]. Recently, a consistent framework for segmentation was proposed in the study of Reference [32], which presents a thorough review of possible techniques. However, rapid progress in this discipline meets some important obstacles against spreading into aiding of STEM images and EDS compositional maps analysis. One obstacle is that ML-based approaches require large sets of labeled training data, while, in conventional STEM-EDS imaging, only several SI can be collected within a reasonable period of experimental time, mainly due to relatively low efficiency for X-ray signal collection. Examples of successful segmentation of images are presented by Uusimaeki et al. [33]. The software called AutoEM is presented with some examples of particles identifying from TEM images. Several well-established image processing methods (for filtering and segmentation) are used. Excellent results were obtained; however, it has to be emphasized that the input images had also excellent quality (very high contrast and noise-to-signal ratio). In daily routine, STEM-EDS images have much worse quality, and such traditional approaches, based on filtering and thresholding, suffer from very low signal-to-noise ratio.

An alternative approach is based on employing primary additional knowledge, rather than only image processing algorithms. One feasible solution is an image decomposing on the ground of local features. Jany et al. [34] proposed application of FFT within local window moving over a micrograph and later application of blind decomposition of the data via Non-Negative Matrix Factorization. Their approach is sensitive to the local features of the image such as symmetry, orientation, and characteristic spacing. It is worth mentioning is that the method does not require a learning process. The other approach to include a domain-knowledge was presented in Reference [35], based on including an expert knowledge in the algorithm. Mirzaei and Rafsanjani [36] presented an application of image processing methods, including nonlinear denoising, edge sharpening filtering, and segmentation to identify nanoparticles from TEM images. The algorithm is dedicated for particles with circular cross-section. With this assumption, computational efficiency and particles’ identification quality were significantly improved, at the expense of loss of generality.

The common drawback of all methods presented above is neglecting a part of initially available specimen knowledge, e.g., chemical composition and repartition of elements. Combination of STEM imaging with compositional imaging by simultaneous acquisition of EDS signal gives the possibility to use information obtained from these complementary methods. Mevenkamp et al. [20] applied this approach into filtering of an image, but only in the context of NLM filtering. Furthermore, a synthetic EDS dataset was used. In this paper, we present an approach extending this concept and applying a multispectral signal analysis for the whole process of automated recognition of cellular structure from denoising to segmentation.

### 1.5. Multispectral Images

If data includes more than one type of information (multi- or hyperspectral images), the problem of distinguishing specific regions of a sample is a clustering problem. There are several approaches available, such as Gaussian mixture clustering [37], SVM [38], the k-means [35], or decision-tree [39]. Very recently, Georget et al. presented a paper focused on identifying phases with combination of BSE and SEM-EDS images [40]. They presented a comprehensive approach to process multispectral data consist of denoising, researcher-defined clustering and visualization. The results are very promising; however, a share of a manual work is significant.

Automation of phases identification with SEM images is a core of Automated Mineralogy—a range of analytical solutions for quantitative analysis of minerals, rocks, and ceramic materials. Schulz et al. presented applied summation of low-counts spectra characteristic for predefined minerals to increase image quality. It has been proved that this approach is very promising for investigated rare earth element ores [41]. Iglesias et al. presented an application of image processing algorithms for automated identification of phases in iron ores. A complex scripts were applied, with several manually fitted coefficients [42]. Presented results are promising; however, it should be emphasized that the processed samples were relatively easier to analyze than typical STEM images of additively manufactured Inconel 625 samples. Moreover, effectiveness of the method was presented only for limited number of cases

### 1.6. Metal Alloys and Additive Manufacturing

Automatization of microscopic images of additively manufactured metal alloys are rather rare. Snow et al. presented an application of Neural Network for detecting flaws in layerwise images of a build. Analyzed images were acquired with X-ray computed tomography (XCT). Obtained accuracy is high but, as other NN-based works, large number of labeled training examples were necessary [43]. 

Applications of ML for analyzing of microstructures of additively manufactured materials are reviewed by Wang et al. [44]. Several important works are discussed, but the most important conclusion is that character ML algorithms require labeled data or, in the case of algorithms with ability to extract representative features automatically, large number of training images (thousands). Johnson et al. presented another wide review of applications of ML in image analysis application in AM materials [45]. Examples include classification of grain structures, measurements of grain size, pore size calculations, and more. Miyazaki et al. [46] presented an application of a complex set of ML techniques of image processing to classify a and b phases in Ti–6Al–4V AM alloy, basing on SEM data. The presented results confirm ability to automatically distinguish these two phases.

Gupta et al. [47] presented an approach to identify phases in steels, based on SEM images processing. Local binary pattern, random forest, and Otsu operators are used for extracting features, classification, and segmentation of microstructure images. The SEM images had been denoised with Gabor filter with coefficient manually fitted. Denoising had been followed with histogram equalization. The Local Binary Pattern algorithm had been used to describe 72 SEM microstructures with a set of properties, which, in turn, had been used to develop the classifier by random forest technique. All 72 images had been manually labeled prior to the process. The images had been segmented with Otsu algorithm, and distinguished phases had been assigned to predefined ones on the ground of the type of steel. The presented results confirm high accuracy of quantifying phases fractions.

From this short state-of the-art review, several conclusions might be drawn. A lot of efforts are put on denoising, segmentation, and phase identification with ML techniques. However, the need of labeled data or abundant training sets limits their practical applications. Classical denoising and segmentation algorithms might be very effective, as well, but they generally require a significant manual fitting to particular conditions and materials. Multispectral analysis is an efficient tool in automatized processing of data, including EDS images, but its applications in identification of phases in metal alloys are very rare. Hence, the efficient at highly automatized way to identify phases in metal alloys, particularly additively manufactured samples, must be still investigated.

Phases identification process in its assumptions is straightforward. The measurements (usually an electron microscopy analysis) provide data, which is processed to maximize information gain. None of the currently available techniques can identify phases directly. Hence, additional step of data interpretation is necessary. The interpretation might be generally based on three approaches: (i) researcher’s intuition and experience, (ii) computational modeling, and (iii) comparative analysis. Currently, the researcher’s interpretation is inevitable—as we showed in the literature review, none of currently available methods can specify the microstructure fully automatically. On the other hand, raw measurements are far too complex to be directly interpreted by human. The goal of the presented research has been not to fully automatize the process of phase identification but to design a solution, maximally decreasing the amount of human-researcher work while keeping reliability on a such high level as it is possible. Hence, if human researcher assistance might ensure high reliability with minimal involvement required, it was decided to not replace them with a fully automatic one.

Another approach to phase analysis is to perform thermodynamic modeling for the average chemical compositions of automatically detected particles, that show the theoretically possible phases in equilibrium conditions. This allows for the creation of a reference system for real results obtained by experimental method. In addition, it enables determination of the influence of temperature on the occurrence of individual equilibrium phases. This information is valuable when interpreting the evolution of identified phases during high temperature exposure. 

Therefore, the aim of this work is to demonstrate the novel approach for automated recognition of precipitates in compositional EDS maps and to verify the results with the microstructural analysis and thermodynamic modeling. 

## 2. Material and Methods

### 2.1. Material

Specimens of additively manufactured Inconel 625 were delivered by Bibus Menos Sp. z o.o. (Gdańsk, Poland). The chemical composition of the alloy is given in Table 1. Specimens fabricated by Direct Metal Laser Sintering^®^ (DMLS), the commercially available L-PBF process of EOS GmbH (Krailling, Germany), were post-build annealed at a temperature of 980 °C for 1 h and slowly cooled in an argon atmosphere. Detailed characterization of the microstructure in the stress-relieved condition can be found in Reference [7]. Subsequently, the specimens were isothermally annealed at a temperature of 800 °C for 5, 100, and 500 h, and cooled in air. This temperature was selected because it is typical for the use of Inconel 625 for prolonged exposure in aerospace, and energy industry applications. 

### 2.2. Microstructural Analysis

The TEM microstructural investigation was performed using JEM-2010 ARP microscope (Jeol, Tokio, Japan). Thin foils for TEM were prepared by electropolishing. STEM investigation was carried out using a Tecnai Osiris microscope (FEI, Hillsboro, OR, USA) operating at 200 kV, equipped with a ChemiSTEM^TM^ system. STEM imaging was performed by applying the high angle annular dark-field (HAADF) mode. Microanalysis of the chemical composition was performed using energy-dispersive X-ray spectrometry (EDS) with the use of Esprit (Bruker, Billerica, MA, USA) software. The distribution of the specific chemical elements was determined using STEM images coupled with EDS maps. The beam current was low to avoid sample damage; thus, the acquisition time was long enough to achieve the signal-to-background ratio minimum of 3:1 for every peak/pixel in the EDS spectrum selected for the collection of elemental maps. The EDS spectra contained Cu peaks generated by the copper support grid of the specimens. To exclude the Cu from quantification, this element was chosen for deconvolution only and was not included in quantitative analysis, as well as in the subsequent automatic detection of microstructural features. The standardless Cliff-Lorimer quantification model was used, as it is convenient for quantitative EDS measurements of thin specimens with relatively high accuracy. EDS spectra recorded at every pixel in the elemental maps allowed us to reconstruct the quantitative EDS maps and linescans, which were used to determine the mean concentrations of chemical elements in the analyzed phases. The indexing of the electron diffraction patterns was performed with the use of JEMS software (version 4.4131U2016, JEMS-SWISS, Jongny, Switzerland) by P. Stadelmann [48]. The crystallographic data of analyzed phases available in Atom Work software (NIMS, Tsukuba, Japan) [49] were used.

### 2.3. Thermodynamic Modeling 

The equilibrium phase stability of the Inconel 625 was calculated using the FactSage™ 7.2 program in conjunction with the SGTE 2017. The calculations were carried out in the temperature range 200–1000 °C using a 10 °C temperature step and the alloy nominal composition given in Table 1. Moreover, the relative amounts of equilibrium phases that can exist at a temperature of 800 °C for the measured compositions of the matrix and precipitates were calculated. 

### 2.4. Automatic Detection of Precipitates

The first part of the proposed approach is processing measurements. In our research, the raw data are STEM-EDS data. 

We also assume that the chemical composition of an investigated sample is known with high level of certainty. A classical way of EDS data analysis is based on prior identification of chemical elements. In our work, we choose a different approach. We treat measurements analogously to a data-science approach, without assigning a physical or chemical knowledge to data. Instead, for the majority of the data processing steps, we rely on statistical tools.

EDS data for a single 2D measurements are large 3D datasets. A raw measurement in our case would be arrays of 1024 × 1024 × 4096 integer values representing a number of registered electrons of particular energy (4096 levels) for each ‘pixel’ of analyzed region (1024 × 1024). EDS maps usually suffer from low signal-to-noise ratio, which is caused by an urge to shorten a measurement duration. To maximize information’s gain from data, several operations must be performed—summarization of measurements over domains for particular energy peaks (this step results in generating single, 2D map for each peak) and improving quality of obtained images. While the goal of our research is automatization, identification of energy peaks is performed manually. This is caused by significant disadvantages of automatic peaks’ finding (relatively high probability of incorrectness of solution) and easiness of manual solution—since the chemical composition is known, it is straightforward to define all possible peaks. Other steps, on the other hand, is fully automatized. 

The procedure starts with a raw STEM-EDS datafile readout followed by the EDS peak identification. The raw data of a Spectrum Image (SI) is represented as a 3D matrix of integral values (x, y, E), where (x, y) correspond to a 2D image pixel coordinates, and E consists of a column-by-column EDS spectral information. *HyperSpy* library was used to process EDS data [50]. With the tools provided by this library, energy ranges defined for particular peaks were converted to the ranges of counts (ΔE). For each recognized peak, all counts are summed in particular energy range of ΔE channels (keV) (separately for each (x, y) pixel). A depth of the data structure is reduced from a number of energy channels (4096 in analyzed cases) into a number of recognized peaks, keeping width and height of the peaks unchanged. For each (x, y) pixel of an image, a vector of integral numbers represents the number of X-ray counts in a particular energy range ΔE that corresponds with detected EDS characteristic peak (i) Equation (1).
(1)$$intensities=\left[{\mathrm{intensity}}_{\left(∆E\right)1},{\;\mathrm{intensity}}_{\left(∆E\right)2},\;\ldots {\;\mathrm{intensity}}_{\left(∆E\right)N}\right]\;.$$

The next step deals with a common problem with the thickness of the wedge-shaped sample in STEM, resulted from a thin foil preparation by electropolishing. The wedge shape sample thickness may lead to a different electron beam interaction volume in the sample, thus altering the emerging X-rays and subsequently detected EDS signal intensities. To compensate for the sample thickness correlated EDS peak intensities, a semi-empirical normalization procedure was applied, based on a linear regression. The detailed unwedging algorithm for 2D (x, y) image is presented in Figure 2.

Digitally unwedged images are smoothed with Gaussian filter with a small standard deviation for Gaussian kernel (default value for 1024 × 1024 image is 3) and normalized to the range (0,1). Gaussian filtering was performed with the *gaussian* method from scikit-image library [51]. Next, the correlation matrix between images is computed. First, all data is converted to the form of *DataFrame* from pandas library [52]. Then, the *corr* method from this library is applied. This method returns a correlation matrix. If any of images’ pair have Pearson correlation coefficient above 0.5, the images are summed (a ‘layer’ with index indicated by a row number in correlation matrix is algebraically summed with a ‘layer’ with index indicated by a column number) and renormalized to the range (0,1). Both original layers are removed from data and replaced by the new layer. These two steps are repeated till none of the images’ pairs have Pearson correlation coefficient above 0.5. 

The pre-prepared image is processed with a set of operations. First, the image is binarized with threshold value computed with triangle algorithm [53]. For this purpose, a set of methods provided by *scikit-images.filters* module is used. First, each image is smoothed with gaussian method. Then, the method *thresholding.threshold_triangle* is used. Positive values (“bright pixels”) indicates pixels with a number of counts higher than the threshold. Since the input images are relatively noisy, the result includes numerous defects (small dark patches in generally light areas and opposite). These “small” objects are removed. Some might be an effect of noise, and others might be a real feature, but with size significantly smaller than the size of objects to be identified. The “size” is understood as an area of directly connected pixels with the same value. This step consists of diluting (*morphology.dilute* method), binary closing (*morphology.binary_closing* method), and eroding (*morphology.erosion* method). 

After the presented above operation, the 3D SEM dataset is replaced with a set of binary images, each representing regions with elevated intensity, computed for combinations of correlated maps of intensities for predefined energy ranges. The quality of images is additionally improved due to application of described above graphical filters. Further, all these images are logically summed (OR operator) to represent all regions, indicating ‘uncommon’ composition. For all logical operation on images, NumPy library was used.

Further steps are focused on identification of phases for each ‘uncommon’ region. The first step requires some human-provided knowledge. Original images (before summing correlated ones), believed to represent peaks coming from different bands of a single chemical element, are summed and renormalized. The mask defined with ‘uncommon regions’ indicator is applied (AND logical operator applied on mask and images for chemical components maps). From the obtained images, regions with intensities in upper quartile are identified. Resulting binary images might be described as maps of regions with high share of particular components and being uncommon. Coherence of the images is improved with some image-processing procedures (applying naming after scikit-image numerical library, dilation, binary closing, and erosion).

The last step, based on STEM-EDS data, is an attempt to bind ‘uncommon’ regions with particular phases. Since quantitative analysis with low signal-to-nose ratio is unreliable, we define phases in qualitative mode. A phase is constituted by combination of binary logical values, indicating “increased” or “not increased” share of particular chemical component. Furthermore, “groups of phases” might also be defined. Such a group is defined with trinary logic, as a combination of values indicating “increased”, “not increased”, and “not relevant” shares. The problem of attaching phases to specimen regions might be treated as a multicriteria optimization function. The objective functions depend on number of used phases, number of used groups of shapes, and area of regions not assigned to any phase or groups of phases. Unfortunately, solving such defined problem is complex, and, currently, it is not automatized. Instead, a researcher must fulfill this step with an “educated guess”, supported by his/her materials science knowledge and computational procedure identifying coverage of “uncommon” regions with a given combination of available phases or phases’ groups.

## 3. Results and Discussion

### 3.1. Microstructural Analysis and Thermodynamic Modeling 

Equilibrium phases possible to occur for the overall composition of the alloy given in Table 1 were determined using the thermodynamic calculation by means of FactSage software and SGTE database. Figure 3 shows the graph of equilibrium phases in IN625 alloy as a function of temperature. The graph is hypothetical, assuming that the alloy with the nominal composition is under equilibrium conditions. The equilibrium phases that occur at a temperature above 700 °C are the γ solid solution (FCC_1_#1 and FCC_2_#1), and intermetallic phases σ (SIGM#1), δ (NI_31_), and P (P_PH), as well as M_23_C_6_ carbide (not visible at this scale with the content less than 2 wt.%). These results make it possible to predict the phase composition in Inconel 625 subjected to long-term annealing. 

Figure 4 shows the TEM, STEM-HAADF, and EDS spectral images of the L-PBF Inconel 625 post-build annealed at a temperature of 980 °C for 1 h. The micrograph shows the cellular structure with fine particles at cell wall regions, as well as inside the cells. EDS spectral images, showing distribution maps of constituent chemical elements, reveal the microsegregation of niobium, molybdenum, and silicon to the cell walls. Moreover, it can be seen that nanoparticles with dark contrast due to the content of lighter elements contain aluminum and oxygen, so they are aluminum oxide inclusions. In the further part of the study, inclusions of oxides were also observed in the samples after annealing at 80 °C, but the analysis concerned only precipitates. 

The use of electron diffraction in TEM allowed for phase analysis of such tiny particles. Phase identification of the precipitates showed the presence of the Laves phase particles rich in Ni, Cr, Mo, Nb, and Si. Meanwhile, the particles rich in Nb were identified as MC carbides. This finding is in agreement with literature data which state that, in superalloys, Nb segregates strongly during solidification to the liquid phase and that the Nb-rich MC carbides and the Laves phase precipitates can be formed in eutectic reactions [7]. 

After annealing at a temperature of 800 °C for 5–500 h, the microstructure was significantly modified (Figure 5). Densely distributed plate-like precipitates were formed inside the grains and along grain boundaries. Diffraction analyses revealed that they are precipitates of the Ni_3_Nb-based δ phase. Moreover, the particles with granular morphology identified as the Laves phase contained Mo, Nb, and Si. The exemplary diffraction patterns, together with their solutions for the δ phase and the Laves phase, are given in Figure 5d,e. The SAED pattern along the [001]_γ_ matrix zone axis is superimposed with the [332] pattern of the δ phase (Figure 5d). In turn, the [011]_γ_ pattern in Figure 5e is superimposed with the [131] pattern of the Laves phase. 

Since this work focused on the automatic analysis of the precipitates based on EDS maps, the results of the analysis of the microstructure and chemical composition were used to select groups of precipitates in terms of similar chemical composition. The results for one case, a sample annealed at 800 °C for 500 h, are provided. Figure 6 shows the STEM-HAADF image and EDS maps of the chemical elements of this specimen. Based on EDS maps, four groups of precipitates were distinguished: group 1 containing mainly Ni and Nb (δ phase), and group 2 rich in Mo and Si (Laves phase), as well as the groups of precipitates, which were not identified as different phases by electron diffraction, namely group 3 containing Ti, Ni, and Cr, and group 4 containing Ti, Cr, Ni, and C. Chemical composition of the matrix and particular groups of precipitates determined on the basis of the quantitative EDS microanalysis are given in Table 2. Subsequently, using FactSage, calculations of the mass fraction of hypothetical equilibrium phases that could be formed with such compositions were performed. The results of the thermodynamic modeling are given in Table 3. 

For the compositions of the matrix and distinguished groups of precipitates, the discrepancies between the observed and predicted phases were noticed. In particular, for the composition of matrix, the predicted to be in equilibrium is the γ phase and the ε phase, being the M_2_C carbide, with a minor fraction of graphite. For group 1, which was identified as the δ phase, calculated equilibrium phases are the γ, ε, M_7_C_3_, and graphite. For the measured composition of group 2, identified as the Laves phase, as well as for group 4, the predicted phases are the same as for the composition of the matrix, whereas, for the composition of group 3, the equilibrium phase that is predicted to be present at the highest mass fraction is the Ni_3_ (Ti, Al)-based intermetallic γ′ phase, and the remaining possible phases are γ, ε, and M_23_C_6_. Generally, the results of the thermodynamic modeling confirm that, for compositions given for calculations, a reasonable agreement between existing and predicted phases was only achieved for the matrix. However, such results were expected since the microstructure of the L-PBF Inconel 625 was far from equilibrium, and such a state was not achieved after annealing at 800 °C for 500 h. Another reason for the differences in the thermodynamically observed and predicted phase composition may be the fact that the chemical composition was determined by means of STEM-EDS using specimens in the form of thin foils, in which the precipitates are surrounded by a matrix. This causes inaccuracy in determining the composition; therefore, the method used does not allow for automatic detection and recognition of phases. The STEM-EDS method was purposefully selected to obtain information about the chemical composition in the form of a dataset from the analyzed area of the sample by means of automated measurement. However, the results showed that the accuracy is unsatisfactory, and, to obtain greater precision, EDS point analysis should be performed. However, in many EDS setups, this method is not automatic and is performed interactively in the areas selected by the researcher while performing the experiment. To be able to extract the EDS point analysis during the data post-processing, it has to be acquired in a Spectrum Imaging (SI) mode at the first place. 

### 3.2. Application of the Elaborated Algorithm for the Automatic Detection of Precipitates in L-PBF Inconel 625

The algorithm presented in the Section 2.4 was verified with analysis of L-PBF Inconel 625 sample annealed at 800 °C for 500 h. STEM-HAADF image of the microstructure together with acquired EDS sum spectrum (number of X-ray counts summed over all 2D image pixels), and cropped EDS spectrum with background removed are shown in Figure 7. Defined peaks are ‘OKα’, ‘NiLα’, ‘CrKα’, ‘CrKβ’, ‘NiKα’, ‘NiKβ’, ‘NbLα’, ‘MoLα’, ‘NbKα’, ‘MoKα’, ‘SiKα’, ‘TiKα’, ‘AlKα’, ‘NKα’, with energy ranges, respectively, (0.3759, 0.676), (0.7015, 1.0015), (5.2647, 5.5647), (5.7967, 6.0967), (7.3109, 7.6109), (8.1146, 8.4146), (2.07, 2.25), (2.25, 2.55), (16.465, 16.7650), (17.330, 17.63), (1.59, 1.89), (4.362, 4.662), (1.33, 1.63), (0.292, 0.402) (all values in keV).

Three-dimensional results of EDS are flattened with summing in ranges connected to particular peaks, transforming a 1024 × 1024 × 4096 matrix to a 1024 × 1024 × 14 one. Exemplary results are shown in Figure 8. Further, correction of wedging and Gaussian smoothing are applied. Results are shown in Figure 9. In the next step, correlation table is computed for the data treated as 14 separate images. As it was described in the previous section, images with correlation coefficient above 0.5 are summed, and these steps are repeated till none of the coefficients are above this threshold. The first and last correlation tables are shown in Table 4.

From this step, images have no binding to real physical values; they must be treated as a form of abstract representation of inhomogeneity of a sample microstructure. Normalized results are shown in Figure 10. Identifying of regions, where values are within the fourth quartile for any of images (7 in the presented case), and a map of ‘distinctive’ regions is obtained (shown in Figure 11). These regions are compared to images showing intensities for chemical components, present in the analyzed sample (if more than one spectrum is measured for a particular component, images are summarized). Exemplary images showing increased values of O, N, Cr, and Al are shown in Figure 12.

As it was discussed in the previous section, automatic procedure for phases identification is still to be developed. Manual investigation led to identifying of 5 phases groups, presented in Table 5 and in Figure 13.

## 4. Summary

The microstructural and compositional analysis by means of TEM, STEM, and EDS were employed to investigate the precipitates in the L-PBF Inconel 625 subjected to high temperature annealing. The thermodynamic modeling was used to calculate the stable phases for compositions of experimentally detected phases. The discrepancy between the predicted and observed phases showed that the microstructure of the L-PBF Inconel 625 after annealing at 800 °C for 500 h is far from equilibrium conditions. 

The application of the elaborated algorithm for the automatic detection of precipitates in L-PBF Inconel 625 provides results clearly worse than those which might be provided by a human researcher. However, the goal of the presented work was not to replace a human but to significantly decrease the amount of work needed to analyze images. The presented algorithm is able to identify interesting (“uncommon”) regions within minutes (on a personal computer), proving digital masks of those domains. From that point, a researcher might start a more thorough analysis. Analysis of a coverage of precipitates with manually defined phases’ groups (presented above) is one of the possible paths. Other choice is manual fitting of the algorithm’s parameters to improve the segmentation quality. 

## Figures and Tables

**Figure 1 materials-14-04507-f001:**
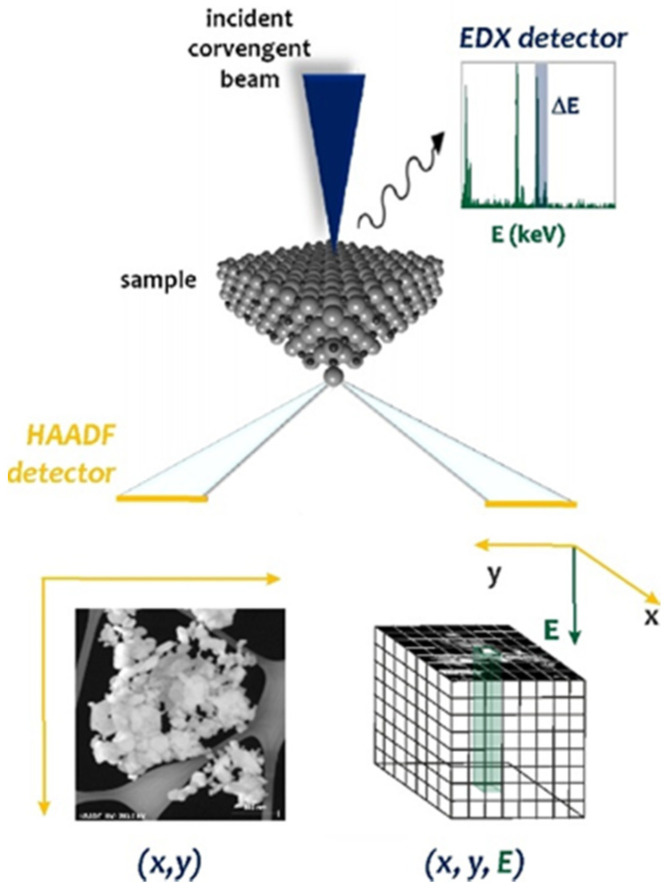
Schematic layout of the Spectrum Imaging (SI) method in STEM mode. The STEM probe is rastered over the specimen (x, y), and local information is retrieved by mapping the HAADF and EDX signals against to the position of the probe (x, y, E). The EDX spectrum shows the characteristic EDX peaks above the background signal. The obtained STEM-HAADF structural images can be complemented by elemental information extracted slice-by-slice, while sampling the EDS spectrum energy space I with an energy corresponding to the ΔE range of the EDS characteristic peaks.

**Figure 2 materials-14-04507-f002:**
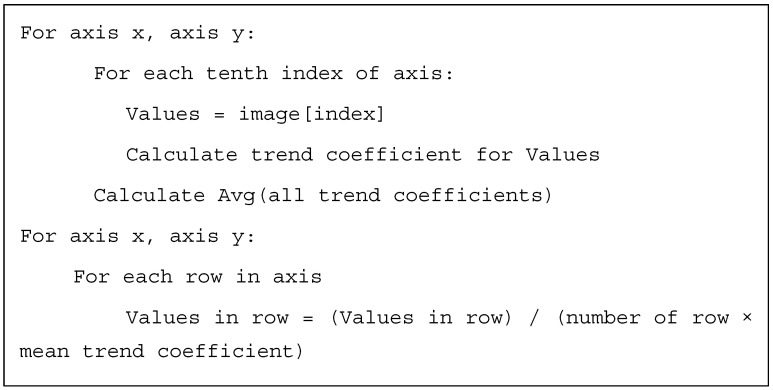
The unwedging algorithm for 2D (x, y) image.

**Figure 3 materials-14-04507-f003:**
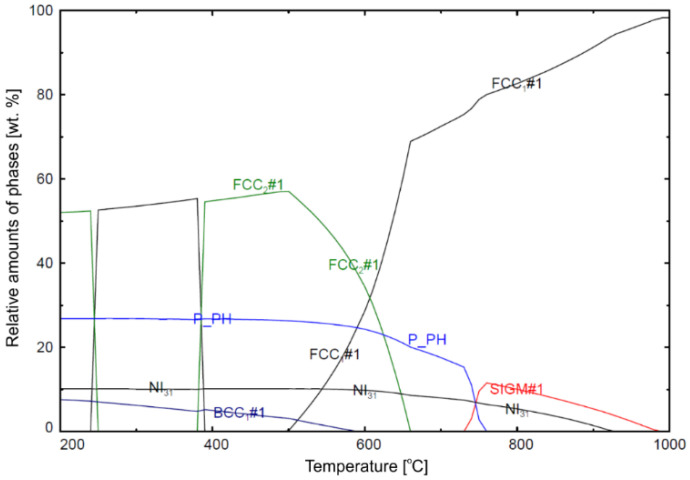
Calculated relative amounts of equilibrium phases in Inconel 625 alloy as a function of temperature.

**Figure 4 materials-14-04507-f004:**
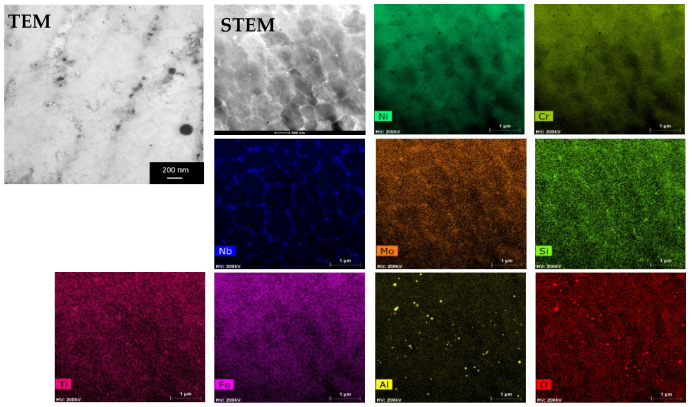
TEM, STEM-HAADF, and EDS spectral images showing maps of constituent chemical elements in the L-PBF Inconel 625 stress-relieve annealed at a temperature of 980 °C for 1 h.

**Figure 5 materials-14-04507-f005:**
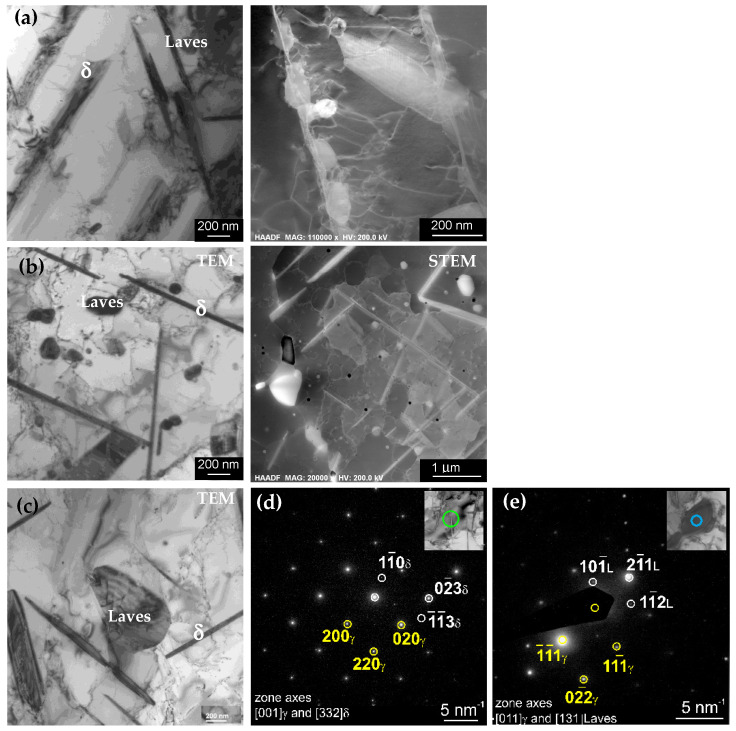
TEM and STEM-HAADF micrographs of the L-PBF Inconel 625 stress-relieved and subsequently annealed at (**a**) 800 °c for 5 h, (**b**) 800 °C for 100 h, and (**c**) 800 °C for 500 h, (**d**,**e**) SAED patterns of the exemplary δ and Laves phase particles shown in the inserts, together with their solutions; the areas covered by the selective aperture are marked by circles.

**Figure 6 materials-14-04507-f006:**
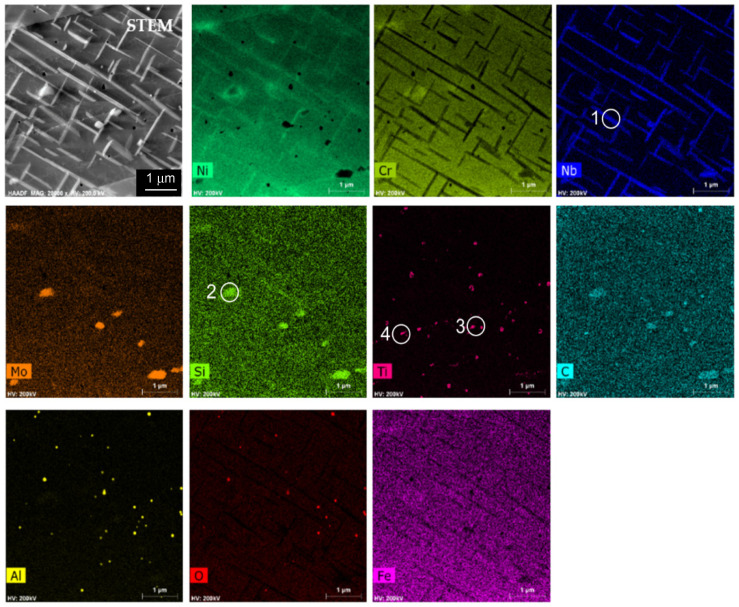
STEM-HAADF image together with EDS maps of chemical elements distribution in the L-PBF Inconel 625 stress-relieved and subsequently annealed at 800 °C for 500 h; exemplary precipitates of groups 1–4 are marked.

**Figure 7 materials-14-04507-f007:**
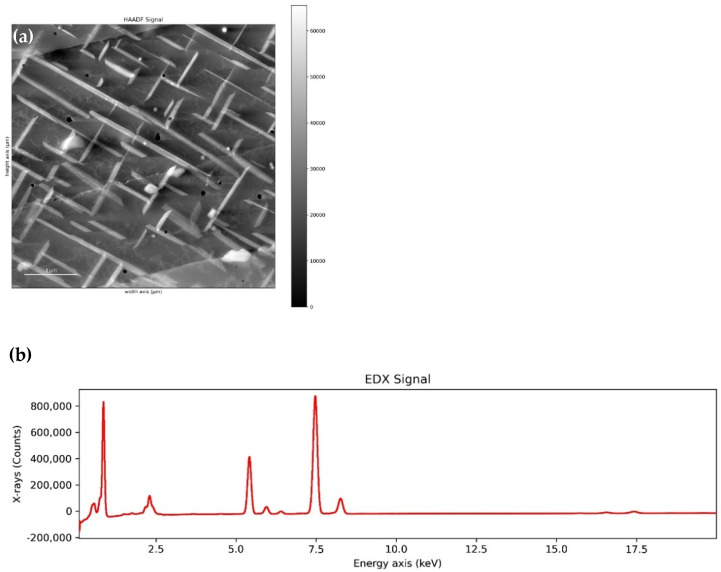
STEM-HAADF image of a microstructure of the L-PBF Inconel 625 post-build annealed at a temperature of 800 °C for 500 h (**a**), and EDS sum spectrum with removed background (**b**).

**Figure 8 materials-14-04507-f008:**
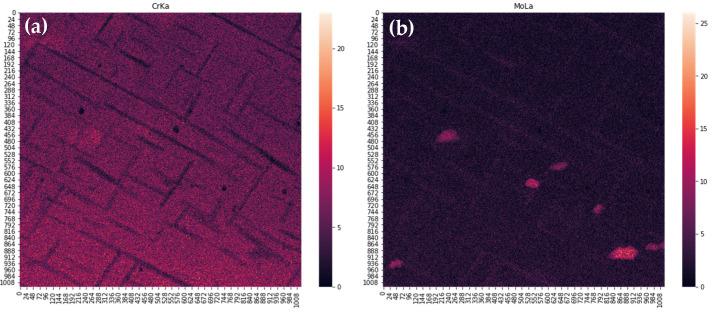
Unprocessed intensities summed over CrKα (**a**) and MoLα (**b**) peaks.

**Figure 9 materials-14-04507-f009:**
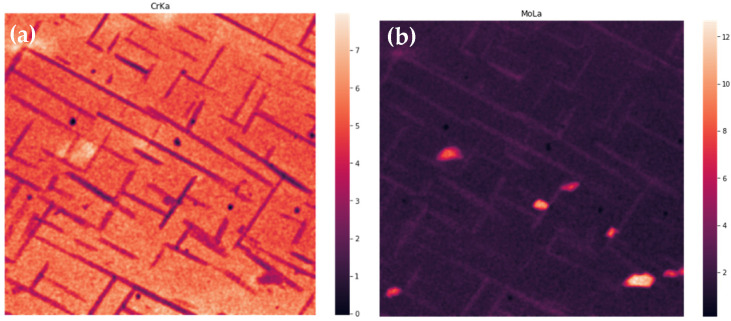
Unwedged and smoothed intensities summed over CrKα (**a**) and MoLα (**b**) peaks.

**Figure 10 materials-14-04507-f010:**
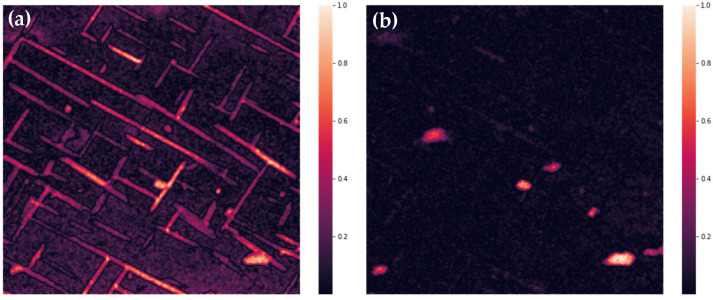
Summed and normalized intensities for images 3rd (**a**) and 4th (**b**) of final set of images.

**Figure 11 materials-14-04507-f011:**
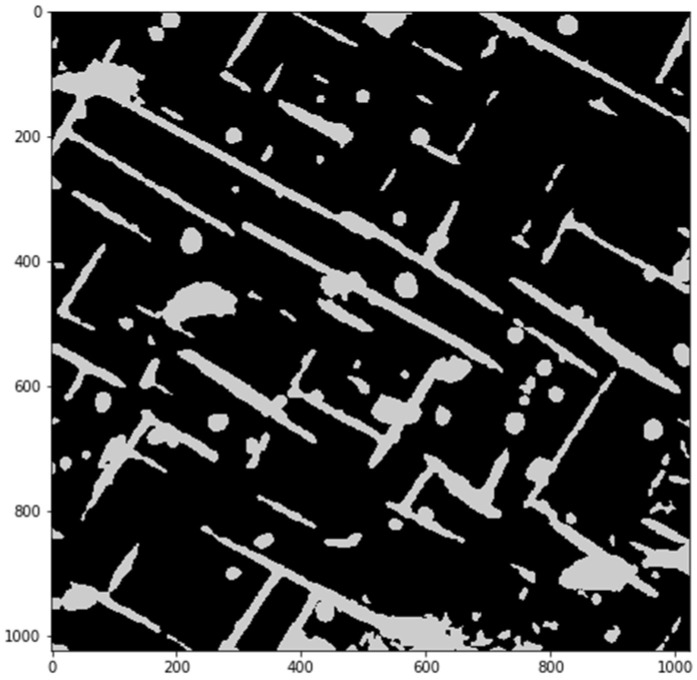
Regions identified as “uncommon”.

**Figure 12 materials-14-04507-f012:**
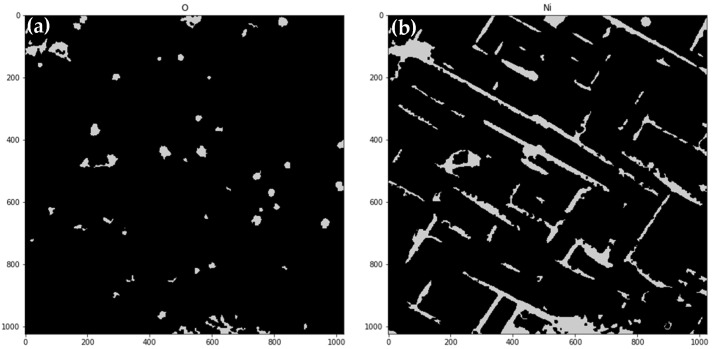
Regions identified as “uncommon” with components share in upper quartile, respectively, O (**a**), Ni (**b**), Cr (**c**) and Al (**d**).

**Figure 13 materials-14-04507-f013:**
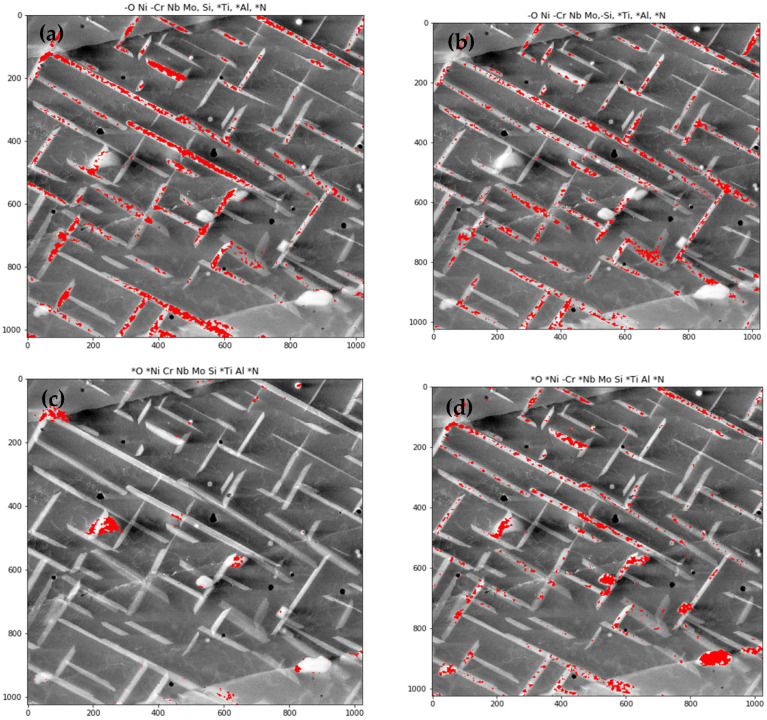
The overlay images of STEM-HAADF and regions identified as phases groups 1–5 marked in red (**a**–**e**), and regions identified as any phase group marked in violet-blue (**f**).

**Table 1 materials-14-04507-t001:** Nominal chemical composition of the Inconel 625 (wt. %).

Ni	Cr	Fe	Mo	Nb	C	Mn	Si	Al	Ti	Co
58.00	22.00	5.00	9.00	3.50	0.10	0.40	0.40	0.30	0.30	1.00

**Table 2 materials-14-04507-t002:** Average chemical composition ± *standard deviation* (in wt. %) of the matrix and designated groups of precipitates in L-PBF Inconel 625 annealed at 800 °C for 500 h.

	Average Chemical Composition ± *Standard Deviation* (in wt. %)											
	O	C	Fe	Ti	Nb	Al	Cr	Mo	Ni	Si	Mn	Co
matrix	0.2	5.2	1.2	0.1	1.4	0.2	22.4	7.8	60.8	0.4	0.2	0.1
	*0.2*	*5.8*	*0.4*	*0.1*	*1.0*	*0.2*	*1.5*	*2.0*	*4.3*	*0.2*	*0.2*	*0.1*
Group 1 Ni, Nb	0.4	10.9	1.1	0.4	5.3	0.4	18.8	5.6	55.9	0.3	0.5	0.4
	*0.4*	*10.0*	*1.0*	*0.4*	*4.2*	*0.4*	*5.5*	*4.6*	*8.1*	*0.3*	*0.5*	*0.4*
Group 2 Mo, Si	0.2	9.3	0.8	0.5	6.5	0.2	17.2	19.2	44.7	0.9	0.4	0.1
	*0.2*	*5.2*	*0.4*	*0.5*	*5.2*	*0.2*	*3.8*	*12.4*	*14.1*	*0.7*	*0.4*	*0.1*
Group 3 Ti, Ni, Cr	1.6	3.2	0.7	5.4	7.9	1.8	15.9	12	50.6	0.7	0.1	0.1
	*1.6*	*3.0*	*0.4*	*2.7*	*3.0*	*1.8*	*3.0*	*9.2*	*6.6*	*0.4*	*0.1*	*0.1*
Group 4 Ti, Cr, Ni, C	0.3	11.3	0.9	3.1	3.4	0.3	17.6	7.5	54.2	0.3	0.2	0.9
	*0.3*	*11.2*	*0.9*	*3.0*	*3.2*	*0.3*	*2.6*	*5.1*	*7.3*	*0.3*	*0.2*	*0.9*

**Table 3 materials-14-04507-t003:** Calculated relative amounts of phases that can be formed at equilibrium conditions at temperature 800 °C for the chemical compositions of the matrix and designated groups of precipitates in the specimen annealed at 800 °C for 500 h given in Table 2.

SGTE Notation	FCC A1#1	FCC A1#2	HCP A3#1	FCC L12#1	M23C6	M7C3	C_ Graphite
phase	γ phase		ε phase	γ′ phase	M_23_C_6_	M_7_C_3_	graphite
**Relative amount (wt. %)**							
Matrix	68.2		29.4				2.4
Group 1 Ni, Nb	63.4	5.7	19.4			3.5	8.0
Group 2 Mo, Si	49.5	9.0	36.3				5.2
Group 3 Ti, Ni, Cr	23.2	15.8	10.8	45.5	4.7		
Group 4 Ti, Cr, Ni, C	61.7	10.5	20.1				7.7

**Table 4 materials-14-04507-t004:** Correlation tables for initial (**a**) and final (**b**) set of images.

(**a**)													
1.000	−0.070	0.439	0.324	−0.024	−0.014	−0.470	−0.271	−0.455	−0.154	−0.168	0.120	0.477	0.297
−0.070	1.000	0.071	0.103	0.711	0.561	0.066	−0.323	0.124	−0.350	−0.058	−0.305	−0.225	−0.146
0.439	0.071	1.000	0.709	0.214	0.170	−0.722	−0.243	−0.745	−0.052	−0.172	−0.235	−0.145	0.016
0.324	0.103	0.709	1.000	0.231	0.184	−0.515	−0.160	−0.535	−0.018	−0.123	−0.207	−0.109	−0.001
−0.024	0.711	0.214	0.231	1.000	0.720	0.131	−0.151	0.175	−0.177	0.051	−0.251	−0.191	−0.097
−0.014	0.561	0.170	0.184	0.720	1.000	0.120	−0.101	0.153	−0.124	0.050	−0.189	−0.141	−0.071
−0.470	0.066	−0.722	−0.515	0.131	0.120	1.000	0.642	0.906	0.417	0.468	0.181	0.093	0.005
−0.271	−0.323	−0.243	−0.160	−0.151	−0.101	0.642	1.000	0.490	0.833	0.587	0.069	0.054	0.053
−0.455	0.124	−0.745	−0.535	0.175	0.153	0.906	0.490	1.000	0.274	0.379	0.177	0.086	−0.014
−0.154	−0.350	−0.052	−0.018	−0.177	−0.124	0.417	0.833	0.274	1.000	0.495	0.035	0.035	0.064
−0.168	−0.058	−0.172	−0.123	0.051	0.050	0.468	0.587	0.379	0.495	1.000	0.024	0.071	0.048
0.120	−0.305	−0.235	−0.207	−0.251	−0.189	0.181	0.069	0.177	0.035	0.024	1.000	0.386	0.355
0.477	−0.225	−0.145	−0.109	−0.191	−0.141	0.093	0.054	0.086	0.035	0.071	0.386	1.000	0.302
0.297	−0.146	0.016	−0.001	−0.097	−0.071	0.005	0.053	−0.014	0.064	0.048	0.355	0.302	1.000
(**b**)													
1.000		−0.041		0.484		−0.226		0.120		0.477		0.297	
−0.041		1.000		0.025		−0.156		−0.281		−0.211		−0.119	
0.484		0.025		1.000		−0.386		−0.229		−0.125		0.006	
−0.226		−0.156		−0.386		1.000		0.048		0.063		0.063	
0.120		−0.281		−0.229		0.048		1.000		0.386		0.355	
0.477		−0.211		−0.125		0.063		0.386		1.000		0.302	
0.297		−0.119		0.006		0.063		0.355		0.302		1.000	

**Table 5 materials-14-04507-t005:** Phases groups. Sign ‘-’ indicates “not increased”, no sign indicates “increased”, and ‘*’ sign indicates “not relevant”.

Phase 1	-O	Ni	-Cr	Nb	Mo	Si	*Ti	*Al	*N
Phase 2	-O	Ni	-Cr	Nb	Mo	-Si	*Ti	*Al	*N
Phase 3	*O	*Ni	Cr	Nb	Mo	Si	*Ti	Al	*N
Phase 4	*O	*Ni	-Cr	*Nb	Mo	Si	*Ti	Al	*N
Phase 5	O	-Ni	-Cr	*Nb	-Mo	-Si	Ti	Al	N

## Data Availability

The data presented in this study are available at the Corresponding author.

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
