# Peer review of "Towards Automatic Detection of Precipitates in Inconel 625 Superalloy Additively Manufactured by the L-PBF Method"

_materials, 2021, doi:10.3390/ma14164507_

Round 1

Reviewer 1 Report

Dear authors,

Thank you very much for the interesting article. This is an interesting work on the post-processing of the precipitate identification techniques in superalloys. The review of literature in the introduction section is well structured. The evaluation of the experiments has been carefully taken. Although the results of the developed algorithm have been presented and discussed, however the algorithm by itself and in practice, has not been addressed well to be applicable by the readers which is the main missing part in the presented manuscript. Please have a look at my comments as follows:

  1. Line 270: “The acquisition time was long enough to achieve the signal-to-background ratio minimum of 3:1 for every peak/pixel in the EDS spectrum selected for the collection of elemental maps”.
  • How the criteria of 3 times higher peak/pixel ratio has been selected? What have been the variables to tune this ratio and how each of them affects that?

  1. Line 272: “The EDS spectra contained Cu peaks generated by the copper support grid of the specimens, which were not included in the subsequent automatic detection of microstructural features
  • How the effect of Cu in quantifications have been corrected? We are all aware of the fact that ignoring a peak which is present in the spectrum, affects the total unnormalized percentage. How much is the difference between total percentage in normalized and unnormalized quantification?

  1. When a quantification in EDS analysis is being used at this level, the deconvolution settings, the quantification model and of course the reason of such selections can be reported.

  1. Which chemical composition have been used in FactSage “equilibrium phase stability” calculations? The presented composition in table 1 seems to be the nominal composition. However, the actual composition might be different specially during the LPBF additive manufacturing technique. Furthermore, this technique imposes metallic or carbon impurities during the fabrication process. The later fact has also been interpreted by authors in the manuscript (Line 512). How the current thermodynamic calculation has been validated for the actual composition?

  1. The fluorescence effect causes overestimation and underestimation of the concentrations. How the fluorescence effect has been corrected? If it’s been performed, the correction factors can be reported.

  1. In the line 264, it’s written that “Thin foils for TEM were prepared by electropolishing”. However, in line 313 it’s been mentioned that the “Focused Ion Beam technique” has been used for the sample preparation. It needs clarification.

  1. The explanation about the Figure 5 is very concise (Line 398). The approach (lattice parameter calculations, crystallographic relationship with the matrix, …) can be presented and be compared to the existing reported data to show how it’s been yielded to the conclusion that the shown phases are d and Laves. Furthermore, the acquisition area for the SAED patterns can be illustrated on the image.

  1. The standard deviation in Table 2 (line 421) supposed to be reported but it’s not clear where it has been included. For each group, there are two reported rows of percentages.
  • a) Are they the maximum and minimum values?
  • b) If it’s so, how it can be explained that in the case of some elements (Ti, Al, Mn, Co in the matrix, Ti, Al, Si Mn, Co in group 1 and etc.) the two values are identical?
  • c) how many images/particles have been studied in this analysis?

  1. The extremely localized compositions have been employed for the thermodynamic calculation (FactSage) to estimate the dominant phases. The reliability of this method in materials science is of question, since the fundamental of the thermodynamic phase stability calculations is on the basis of the phases’ activity and consequently the free energies. However, the activity contribution of the other elements which were existing in the matrix in reality but have not been detected by EDS analysis or were not present at the analysis area, have been eliminated in this methodology. Elucidation is needed by the authors.

  1. How the presence of graphite(!) can be explained in an Inconel alloy (line 429) and 2.4 wt.% (table 3)?

  1. Some typo corrections are needed:
  • 152: “however it have to be emphasized”: has to..
  • 324: “If any if images’
  • 448: “set ups
  • a” can be replaced with alpha (in subscript) in 458-462 and the rest of manuscript (Although it’s common, but it’s less accurate.)
  • 473: “the coefficients is”: are..
  • 486: d “(shown in Figure 11.”: missing “)”

Reviewer 2 Report

Very good work.

I think that manuscript can be accepted in present form. But you can improve state of the art in intoduction. The are several misprints (for example: missed bracket, line 486).

Reviewer 3 Report

Dear Authors,

The presented manuscript tries to address a complex issue relative to the identification of precipitated phases based on the STEM-EDS. In order to do so, an algorithm is developed for the automatic detection of precipitates and identification of interesting regions in the Inconel 625.

The manuscript is easy to read, the methodology used is well described and the conclusion are well supported by the results (thanks for your honesty, because with your work part of the tedious processing of signal can be avoid however human interaction is still needed).

The only comment that I do have is that I would add more details about what is behind the code to postprocessing the images. This way, the reader would be capable of reproducing your work.

Good work!

Round 2

Reviewer 1 Report

Thanks to the authors for clarifications and modifications. Most of the review comments are well addressed and the quality of the manuscript has been improved.

As the last point, I suggest to eliminate the following sentence (Line 321 of V2) if no reference can be added to support the explanation of "generally considered acceptable":

"Such a ratio is generally considered acceptable for the detection limit in analytical procedures."

Warm Regards

Author Response

According to the Reviewer comment, the sentence "Such a ratio is generally considered acceptable for the detection limit in analytical procedures." is removed.